# *In Vitro* Profiling of the Synthetic RNA Polymerase Inhibitor MMV688845 against *Mycobacterium abscessus*

Lea Mann,[a] Uday S. Ganapathy,[b] Rana Abdelaziz,[a] Markus Lang,[a] Matthew D. Zimmerman,[b] Véronique Dartois,[b,c] Thomas Dick,[b,c,d] Adrian Richter[a]

[a]Institut für Pharmazie, Martin-Luther-Universität Halle-Wittenberg, Halle (Saale), Germany
[b]Center for Discovery and Innovation, Hackensack Meridian Health, Nutley, New Jersey, USA
[c]Department of Medical Sciences, Hackensack Meridian School of Medicine, Nutley, New Jersey, USA
[d]Department of Microbiology and Immunology, Georgetown University, Washington, DC, USA

**ABSTRACT** In a library screen of tuberculosis-active compounds for anti-*Mycobacterium abscessus* activity, we previously identified the synthetic phenylalanine amide MMV688845. In *Mycobacterium tuberculosis*, this class was shown to target the RpoB subunit of RNA polymerase, engaging a binding site distinct from that of the rifamycins. Due to its bactericidal activity, rifampicin is a key drug for the treatment of tuberculosis (TB). However, this natural product shows poor potency against *M. abscessus* due to enzymatic modification, and its clinical use is limited. Here, we carried out *in vitro* microbiological profiling of MMV688845 to determine its attractiveness as a substrate for a chemistry optimization project. MMV688845 was broadly active against the *M. abscessus* complex, displayed bactericidal against *M. abscessus in vitro*, and in a macrophage infection model showed additivity with commonly used anti-*M. abscessus* antibiotics and synergy with macrolides. Analyses of spontaneous resistant mutants mapped resistance to RpoB, confirming that MMV688845 has retained its target in *M. abscessus*. Together with its chemical tractability, the presented microbiological profiling reveals MMV688845 as an attractive starting point for hit-to-lead development to improve potency and to identify a lead compound with demonstrated oral *in vivo* efficacy.

**IMPORTANCE** Infections with nontuberculous mycobacteria are an increasing health problem, and only a few new drug classes show activity against these multidrug-resistant bacteria. Due to insufficient therapy options, the development of new drug leads is necessary and should be advanced. The lead compound MMV688845, a substance active against *M. abscessus* complex, was characterized in depth. In various assays, it showed activity against *M. abscessus*, synergy with other antibiotics, and bactericidal effects.

**KEYWORDS** *M. abscessus*, MMV688845, RNA polymerase, drug development, nontuberculous mycobacteria

**M**ycobacterium abscessus is a fast-growing *mycobacterium*, classified as a nontuberculous *mycobacterium* (NTM). It is an opportunistic pathogen that causes serious respiratory infections, especially in patients with preexisting health conditions such as immunosuppression or cystic fibrosis (1). However, infections with *M. abscessus* are not limited to the lungs but can also occur in the skin, soft tissues, and central nervous system (2). Diseases caused by *M. abscessus* are difficult to treat because the pathogen is resistant to numerous classes of antibiotics (3). *M. abscessus* infections often require years of treatment, with 12 months of continuation treatment after sputum conversion with a combination of three or more antibiotics (4). Treatment regimens often include clarithromycin, amikacin, and cefoxitin or imipenem. Despite the combination therapy, cure rates are poor. Acquired and intrinsic resistance, including inducible resistance, causes the loss of efficacy of many antibiotics (5, 6). Like *Mycobacterium tuberculosis*, *M. abscessus* can escape host defense mechanisms by infecting human

**Ad Hoc Peer Reviewer** Sangeeta Tiwari

Address correspondence to Adrian Richter, adrian.richter@pharmazie.uni-halle.de.

The authors declare no conflict of interest.

macrophages, leading to granuloma formation (7). Due to the poor clinical outcomes and the increase of diagnosed NTM infections, the development of new efficacious antibiotics is urgently needed.

MMV688845 ($N\alpha$-2-thiophenoyl-*d*-phenylalanine-2 morpholinoanilide), identified by screening of the Pathogen Box library (Medicines for Malaria Ventures [MMV]) against *M. abscessus* (ATCC 19977) (8–10), is a promising hit compound for drug development focused on NTM and *M. abscessus* in particular. Initially, MMV688845 was discovered as an anti-*M. tuberculosis* (11, 12) hit. The compound shows activity *in vitro* against *M. abscessus* ATCC 19977 in 7H9 medium and in cation-adjusted Mueller-Hinton broth (MHII). Recently, an improvement in the synthetic route for the active R enantiomer was achieved (13). In a previous study (13), the cytotoxicity of MMV688845 was investigated. Cytotoxicity was analyzed against five mammalian cell lines, including A375 (melanoma), HT29 (colon adenocarcinoma), MCF-7 (breast adenocarcinoma), A2780 (ovarian carcinoma), and NIH 3T3 (nonmalignant mouse fibroblast). The compound was evaluated using a sulforhodamine B (SRB) (Kiton-Red S; ABCR GmbH, Karlsruhe, Germany) microculture colorimetric assay in which MMV688845 showed no cytotoxicity (13).

In this study, MMV688845 was characterized in depth as a potent inhibitor against *M. abscessus*. MMV688845 was tested against a range of clinical isolates to demonstrate that MMV688845 is not only effective against selected laboratory strains. To evaluate the bactericidal effects of MMV688845, we performed CFU determinations, both *in vitro* and in a macrophage infection model. For target validation in *M. abscessus*, spontaneous resistant mutants were isolated and sequenced.

To study the synergy of MMV688845 with approved antimycobacterial drugs and to assess its suitability for combination therapy, we performed synergy testing against *M. abscessus* (ATCC 19977) by checkerboard assays. In particular, we were interested in the interaction with macrolides, e.g., clarithromycin, as clarithromycin and rifabutin were shown to be synergistic for *erm41*-positive *M. abscessus* strains. Transcriptional inhibition by rifabutin disables inducible macrolide resistance in *M. abscessus*, inhibiting the expression of *erm41*. Therefore, this study investigated the hypothesis that the RNA polymerase inhibitor MMV688845 exhibits synergy with macrolides (14–16).

## RESULTS

**MMV688845 is active against various *M. abscessus* strains *in vitro*.** To fully determine the potential of MMV688845, we investigated its activity against a panel of *M. abscessus* subspecies and clinical isolates (Table 1). MMV688845 is active against all three *M. abscessus* subspecies and a variety of clinical isolates. Based on the activity of MMV688845 against *M. abscessus* Bamboo and *M. abscessus* ATCC 19977 (Table 1), we investigated the effect of the hit compound against the three subspecies of the *M. abscessus* complex (*M. abscessus* subsp. *abscessus*, *M. abscessus* subsp. *massiliense*, and *M. abscessus* subsp. *bolletii*), which are known to have different antibiotic susceptibilities (17, 18). We found that MMV688845, like clarithromycin and rifabutin, is effective against the reference strains of the three subspecies (Table 1). Against 10 clinical *M. abscessus* isolates, MMV688845 showed activity comparable to the reference strains (15).

To further mimic infections in human macrophages, a macrophage infection model based on THP-1 cells was used to determine the effect of MMV688845 on bacteria growing inside cells, as activity of MMV688845 in a macrophage model has been reported previously (19) (Table 2). We found MMV688845 to be active against intracellular *M. abscessus* with an $MIC_{90}$ of 16 $\mu$M (Table 2).

As demonstrated by Lin et al. (12), analogues closely related to MMV688845 bind to the RpoB subunit of the mycobacterial RNA polymerase at a different site than rifamycins, leading to efficient enzyme inhibition (12, 20, 21). To validate the RNA polymerase as the target of MVV688845 in *M. abscessus*, resistant mutants were isolated (Table 1). For MMV688845-resistant *M. abscessus* strains, there is no cross-resistance to rifamycins,

**TABLE 1** *In vitro* activity of MMV688845 against *M. abscessus* reference strains and clinical isolates

| *M. abscessus* strain | *erm41* sequevar | CLR susceptibility | MIC$_{90}$ ($\mu$M) of: | | |
| --- | --- | --- | --- | --- | --- |
| | | | CLR | RFB | MMV688845 |
| Reference strains[a] | | | | | |
| *M. abscessus* subsp. *abscessus* ATCC 19977 | T28 | Resistant | 1.3 | 1.8 | 7.5 |
| *M. abscessus* subsp. *bolletii* CCUG 50184T | T28 | Resistant | 2.6 | 2.1 | 10 |
| *M. abscessus* subsp. *massiliense* CCUG 48898T | Deletion | Sensitive | 0.3 | 0.8 | 10 |
| Clinical isolates[a] | | | | | |
| *M. abscessus* subsp. *abscessus* Bamboo | C28 | Sensitive | 0.3 | 1.3 | 8 |
| *M. abscessus* subsp. *abscessus* K21 | C28 | Sensitive | 0.6 | 2.4 | 6.6 |
| *M. abscessus* subsp. *abscessus* M9 | T28 | Resistant | 1.7 | 2.2 | 8.9 |
| *M. abscessus* subsp. *abscessus* M199 | T28 | Resistant | 3.4 | 1.5 | 8.6 |
| *M. abscessus* subsp. *abscessus* M337 | T28 | Resistant | 1.2 | 1.2 | 5.4 |
| *M. abscessus* subsp. *abscessus* M404 | C28 | Sensitive | 0.3 | 1.1 | 6.6 |
| *M. abscessus* subsp. *abscessus* M422 | T28 | Resistant | 0.8 | 1.1 | 8.4 |
| *M. abscessus* subsp. *bolletii* M232 | T28 | Resistant | 1.7 | 1.2 | 6.9 |
| *M. abscessus* subsp. *bolletii* M506 | C28 | Sensitive | 0.3 | 1.1 | 4.5 |
| *M. abscessus* subsp. *massiliense* M111 | Deletion | Sensitive | 0.3 | 2.3 | 8.4 |
| *M. abscessus* subsp. *abscessus* ATCC 19977 (RFP assay)[b] | | | | | |
| In 7H9 + ADS | T28 | Resistant | 0.4 | 1.2 | 11.9 |
| In MHII | T28 | Resistant | 0.2 | 2.6 | 10.8 |
| In the macrophage infection model | T28 | Resistant | 50 | 5.5 | 15.9 |
| *M. abscessus* subsp. *abscessus* ATCC 19977 RFB-R1[a,c] | T28 | Resistant | 3.5 | >100 | 10 |
| *M. abscessus* subsp. *abscessus* Bamboo 845-2[a,d] | T28 | Resistant | 0.3 | 1.1 | >100 |

[a]MICs were determined in 7H9 via OD$_{600}$ measurement and evaluated via method B.
[b]MICs were determined by RFP measurement and evaluated via method A.
[c]Spontaneous RFB-resistant strain.
[d]Spontaneous MMV688845-resistant strain.

which was investigated by MIC determination of rifabutin (RFB) against *M. abscessus* Bamboo 845-2 (Table 1).

The MIC$_{90}$ values shown in Table 1 were obtained by determining the optical density at 600 nm (OD$_{600}$) or by measuring red fluorescent protein. For the RFP-based method, the *mycobacterium* is transformed with plasmid pTEC27 for RFP tdTomato expression (22). Fluorescence measurement was used for quantification of bacterial growth because this method is sensitive and specific, as has been shown previously (23, 24). Table 1 shows that the results of the fluorescence-based *in vitro* assay are consistent with optical density measurements.

**MMV688845 has bactericidal activity *in vitro* and in a macrophage infection model.** To assess the bactericidal efficacy of MMV688845, CFU counting experiments were conducted, and the results were compared with those for rifabutin as a reference, including the combination of both agents (Table 2).

We found that MMV688845 achieves a significant reduction of colony forming units of *M. abscessus* in vitro. The MBC90 was determined to be 15 $\mu$M (2× MIC90) against the reference strain ATCC 19977. As shown in Fig. 1, a concentration-dependent reduc-

**TABLE 2** MICs and MBCs of MMV688845 and rifabutin against *M. abscessus* subsp. *abscessus* ATCC 19977

| Growth condition | MIC$_{90}$ ($\mu$M) of: | | MBC$_{90}$ ($\mu$M) of: | |
| --- | --- | --- | --- | --- |
| | RFB | MMV688845 | RFB | MMV688845 |
| 7H9 + ADS | 1.2[a] | 7.5[a] | 2.4 (2 × MIC)[b] | 15 (2 × MIC)[b] |
| Macrophage infection model | 5.5[c] | 15.9[c] | ND[d] | 16 (1 × MIC)[e] |

[a]Determined in 7H9 via OD$_{600}$ measurement and evaluated via method B.
[b]Determined in duplicate via method B.
[c]Determined by fluorescence measurement.
[d]ND, not determined.
[e]Determined in triplicate via method A. For the MBC determination, statistical evaluation was carried out via one-way ANOVA multiple-comparison test, as for Fig. 1.

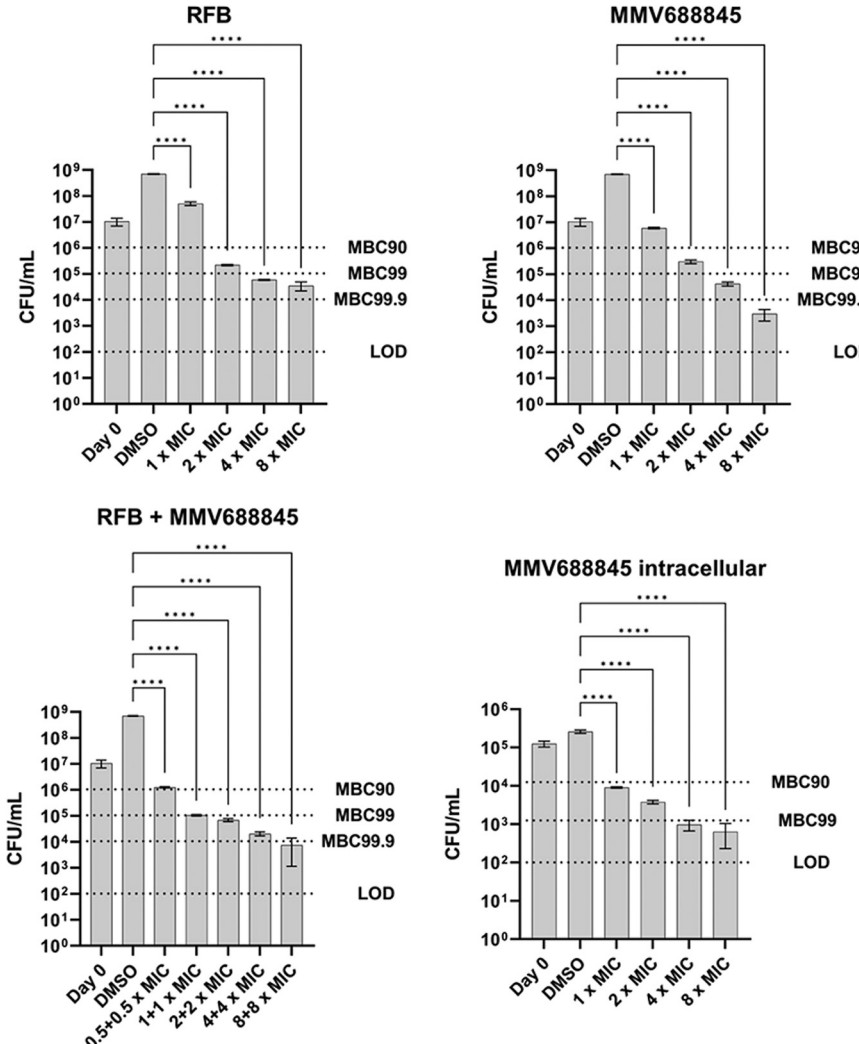

**FIG 1** Bactericidal activity of MMV688845 and rifabutin against *M. abscessus* ATCC 19977. LOD, limit of detection. Experiments were carried out in duplicate or triplicate. Results are means, with the standard deviations displayed as error bars. A one-way analysis of variance (ANOVA) multiple-comparison test was performed using GraphPad Prism 8 software to compare treated groups with the DMSO control. ****, $P \leq 0.0001$.

tion of CFU was observed; however, a reduction of the CFU count by 3 logarithmic units could be achieved only with a concentration of $8 \times$ MIC (Fig. 1). In parallel, the bactericidal activity of MMV688845 was investigated in the macrophage infection model, where a reduction in the bacterial count was also observed and an $MBC_{90}$ of $\leq 16$ $\mu$M ($1 \times$ MIC) was determined. The reference compound rifabutin also showed bactericidal effects with a minimal bactericidal concentration at which 90% of the bacteria are inhibited ($MBC_{90}$) of 2.4 $\mu$M ($2 \times$ MIC) (Table 2).

In combination, a reduction in CFU of about 90% can be achieved with $0.5 \times$ MIC of each drug (Fig. 1), indicating an additive effect and consistent with the observation that rifabutin and MMV688845 engage different binding sites on the RpoB subunit of RNA polymerase (12).

**MMV688845 targets the RNA polymerase in *M. abscessus*.** Data in the literature show that the structural class of phenylalanine amides are inhibitors of the RpoB subunit of the mycobacterial RNA polymerase (12). However, these studies were performed on the RNA polymerase of *M. tuberculosis* with analogues of MMV688845. To confirm the molecular mechanism of action for MMV688845 in *M. abscessus*, six strains of resistant mutants

**TABLE 3** Characterization of *M. abscessus* MMV688845-resistant mutants

| *M. abscessus* subsp. abscessus strain | Batch | $MIC_{90}$ ($\mu M$)[a] of MMV688845 | RpoB mutation(s) |
|---|---|---|---|
| Bamboo | | 8 | |
| Bamboo 845R-1.1 | 1 | >100 | P473L |
| Bamboo 845R-2.1 | 2 | >100 | P473L |
| Bamboo 845R-2.2 | 2 | >100 | G562S |
| Bamboo 845R-2.3 | 2 | >100 | L556P, V557P |
| Bamboo 845R-2.4 | 2 | >100 | Q581R |
| Bamboo 845R-2.5 | 2 | >100 | D576Y |

[a]Determined by measurement of $OD_{600}$ and evaluated via method B. Two batches of spontaneous resistant mutants were generated by selection at 50 $\mu M$ MMV688845. The observed frequency of resistance was $6 \times 10^{-8}$ CFU$^{-1}$.

were isolated and characterized. All resistant strains show a significant increase in MIC (>100 $\mu M$) and mutations in the RpoB subunit of the RNA polymerase (Table 3).

In this study, a frequency of resistance of $6 \times 10^{-8}$ CFU$^{-1}$ was determined. The localization of the above-mentioned mutations leading to MMV688845 resistance was visualized in a homology model of the RpoB subunit of *M. abscessus* RNA polymerase (Fig. 2). The basis for the model is an X-ray crystal structure (PDB code 5UHE) of the enzyme from *M. tuberculosis* which was determined by Lin et al. (12) with an $N\alpha$-aroyl-*N*-aryl-phenylalaninamide (d-AAP 1). Even though d-AAP 1 has small structural differences from MMV688845, it belongs to the same chemical class, which makes a similar binding mode very likely.

The six mutations described in Table 3 are located near the binding pocket of the inhibitor and are highlighted in red in Fig. 2. The fact that the substitution of individual amino acids leads to a loss of efficacy of MMV688845 suggests that the binding site on the RNA polymerase is relevant for the activity of the compound. Glycine 566 (*M. tuberculosis* numbering) is involved in hydrogen bonding with the inhibitor molecule, highlighting its importance in binding. It is likely that in the *M. abscessus* form of the enzyme, glycine 562 takes part in the formation of this hydrogen bond. One of the observed mutations affects this glycine and leads to MMV688845 resistance of the strain.

**Antibiotics for synergy testing with MMV688845.** As a requirement for the synergy testing described below, the individual $MIC_{90}$ values of each antibiotic for *M. abscessus* ATCC 19977 were determined in a microplate dilution assay. We decided to analyze the activity of antibiotics that are used for the treatment of NTM infections *in vitro* under three different assay conditions against *M. abscessus*. For this purpose, two assay media (7H9 and MHII) were used to identify a possible medium dependency of the results. The intracellular MICs were determined in a fluorescence-based macrophage infection model.

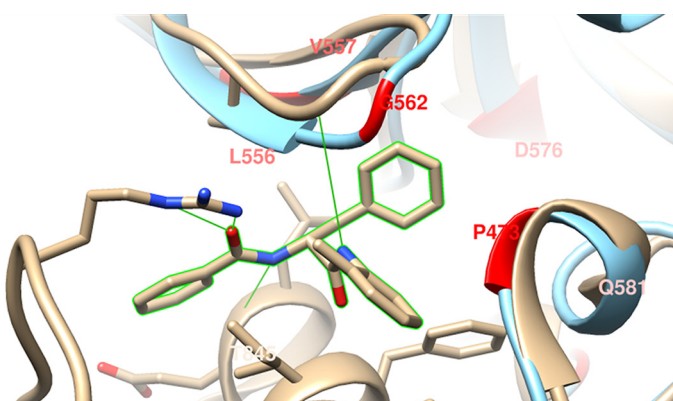

**FIG 2** Visualization of RpoB mutations in an *M. abscessus* homology model. Beige, *M. tuberculosis* RpoB [PDB 5UHE] with d-AAP 1 bound; light blue, *M. abscessus* homology model built using RoseTTAfold; red, *M. abscessus* resistance mutations; green, H bonds. (12, 39).

**TABLE 4** *In vitro* activity of antibiotics against *M. abscessus* ATCC 19977

| Drug[a] | MIC$_{90}$ ($\mu$M)[b] in: | | |
|---|---|---|---|
| | 7H9 | MHII | THP-1 |
| AMK | 1.5 | 2.1 | — |
| BDQ | 0.5 | 1.5 | 0.8 |
| CEF | 36 | 16 | — |
| CLR | 0.4 | 0.2 | 50 |
| AZM | 2.6 | 5.9 | 224[c] |
| EMB | 68 | — | — |
| RMP | 14 | 20 | — |
| TEI | 16[d] | 4.5 | — |
| TIG | 0.8 | 1.7 | 6.1 |
| TLM | 0.8 | 3 | 298[c] |

[a]AMK, amikacin; AZM, azithromycin; BDQ, bedaquiline; CLR, clarithromycin; CEF, cefoxitin; EMB, ethambutol; RFB, rifabutin; RMP, rifampicin; TEI, teicoplanin; TIG, tigecycline; TLM, telithromycin.
[b]Determined by RFP measurement and calculated by method A. —, no growth inhibition was observed.
[c]MIC$_{50}$.
[d]MIC$_{80}$.

For the initial MIC determination, the broth microdilution assay was performed in quadruplicate on each 96-well plate, which was performed in duplicate. The results were averaged and plotted logarithmically with Origin, and the MIC$_{90}$/MIC$_{80}$ and MIC$_{50}$ were determined through graphic evaluation.

The results of the MIC determination of the individual antibiotics are shown in Table 4. We included antibiotics with comparatively high MICs, such as ethambutol and cefoxitin, to analyze potential synergy with MMV688845. The MICs determined in this study do not show a strong dependence on the growth medium used.

Since *M. abscessus* can survive and replicate in macrophages, we were interested in whether it is possible to detect synergy in the macrophage infection model. For this type of *in vitro* assay, bacterial growth is analyzed by high-content microscopy and automated image analysis. For the evaluation of the assay, image data were recorded for the DAPI (4′,6-diamidino-2-phenylindole) channel for cell counting and in the RFP channel for quantification of bacterial growth. Higher MICs in the macrophage assay were expected, as permeability, metabolic (in)stability, and individual properties of the antibiotics might influence the efficacy in macrophages. To analyze synergy with this advanced *in vitro* method, we determined MICs for the antibiotics first in the macrophage infection assay. Amikacin, as an example, shows no efficacy in macrophages due to its polarity and thus low intracellular permeability. Cefoxitin, clarithromycin, rifampicin, and teicoplanin show up to an 10-fold increase in MIC compared to *in vitro* tests in MHII and 7H9 media. In addition, cytotoxic properties of the substances on THP-1 cells are considered by analysis of the cell count after the incubation period. Rifampicin, for example, shows toxic effects on macrophages (cell count reduced by at least 10% compared to the reference according to microscopic evaluation) at concentrations higher than 125 $\mu$M, so the MIC$_{90}$ could not be determined.

**Synergy testing of MMV688845.** The interaction of MMV688845 with the antibiotics mentioned above was analyzed by checkerboard assays and quantified by calculating the fractional inhibitory concentration index (FICI). Synergy testing by checkerboard assays is a method commonly used to test the effectiveness of antibiotic combinations *in vitro* (25, 26).

Within the FICIs obtained from the checkerboard experiments, no antagonism (FICI > 4) of combinations with MMV688845 was observed. Clearly, for all compound combinations, a reduced MIC of both antibiotics was obtained, compared to the original MIC from the single determination (Table 3). In the majority of the combination experiments, an additive effect is observed.

As mentioned above, we did expect synergy of MMV688845 with macrolides. It is known that rifabutin inhibits the transcriptional induction of *erm41* and thereby enhances the effect of macrolides (16). Therefore, a combination with clarithromycin was analyzed, and the effect of the combination with MMV688845 stood out as the most potent

**TABLE 5** FICI values for combinations of MMV688845 with antimycobacterial drugs against *M. abscessus* ATCC 19977[a]

| Drug[b] | 7H9 | | | MHII | | | THP-1 | | |
| | Conc. ($\mu$M) | Conc. of MMV688845 ($\mu$M) | FICI | Conc. ($\mu$M) | Conc. of MMV688845 ($\mu$M) | FICI | Conc. ($\mu$M) | Conc. of MMV688845 ($\mu$M) | FICI |
|---|---|---|---|---|---|---|---|---|---|
| AMK | 0.8 | 1.6 | 0.7 | 1.3 | 6.3 | 1.2 | — | — | — |
| BDQ | 0.3 | 1.6 | 0.8 | 0.2 | 6.3 | 0.7 | 0.1 | 3.1 | 0.6 |
| CEF | 25 | 1.6 | 0.8 | 12.5 | 1.6 | 0.9 | — | — | — |
| CLR | 0.02 | 3.1 | 0.3 | 0.04 | 3.1 | 0.5 | 3.1 | 1.6 | 0.2 |
| AZM | 0.8 | 3.1 | 0.6 | 1.6 | 1.6 | 0.4 | — | — | — |
| TLM | 0.2 | 3.1 | 0.5 | 0.8 | 3.1 | 0.6 | — | — | — |
| EMB | 50 | 1.6 | 0.9 | — | — | — | — | — | — |
| RFB | 0.3 | 6.3 | 0.7 | 1.6 | 3.1 | 0.9 | 5.0 | 0.8 | 1.0 |
| RMP | 7.8 | 0.8 | 0.6 | 4.7 | 3.1 | 0.5 | — | — | — |
| TEI | 3.1 | 1.6 | 0.4 | 1.6 | 6.3 | 0.9 | — | — | — |
| TIG | 0.1 | 6.3 | 0.7 | 0.8 | 0.4 | 0.5 | 3.1 | 0.8 | 0.6 |

[a]All concentrations were determined by RFP signal. —, not determined.
[b]AMK, amikacin; AZM, azithromycin; BDQ, bedaquiline; CEF, cefoxitin; CLR, clarithromycin; EMB, ethambutol; RFB, rifabutin; RMP, rifampicin; TEI, teicoplanin; TIG, tigecycline; TLM, telithromycin.

(FICI $\leq$ 0.5). The lowest FICI was determined in the macrophage infection model, with a value of 0.2 (Table 5), indicating synergism against intracellular *M. abscessus*. A concentration of 3.1 $\mu$M clarithromycin (0.06$\times$MIC) leads to 90% growth inhibition in the presence of 1.6 $\mu$M MMV688845 (10-fold reduced MIC) in the macrophage infection assay (Fig. 3). In 7H9 broth and MHII, FICIs of 0.3 and 0.5 were obtained, confirming that the synergistic effect is independent of the assay medium used.

We corroborated this result by characterization of other macrolide antibiotics in combination. For this experiment, we chose the macrolide azithromycin and the ketolide telithromycin. We maintained our approach and first determined the MIC$_{90}$ of each compound alone, in 7H9, in MHII, and in the macrophage infection model (Table 4). The MIC$_{90}$s obtained for azithromycin and telithromycin in 7H9 and MHII are higher than those of clarithromycin. Our results are consistent with data found by Aziz et al. (21) for the *M. abscessus* strain Bamboo, which confirm the up-to-10-fold-higher inhibitory effect of clarithromycin compared to azithromycin and telithromycin. Growth inhibition of 90% or more by azithromycin and telithromycin was not achieved in the macrophage infection assay (Table 4), and higher concentrations displayed cytotoxicity against the eukaryotic cells. Therefore, we report MIC$_{50}$ values in this study, but for consistency with the previous results, no FICI was calculated. The combination with azithromycin shows additive effects with MMV688845 in 7H9 but a synergistic FICI of 0.4 in MHII media. For the combination assay with telithromycin, comparable results were observed. The ketolide shows synergy with MMV688845 in 7H9 (FICI 0.5) and additive effects in MHII medium (FICI 0.6) (Table 5).

The RNA polymerase inhibitors rifabutin and rifampicin showed predominantly additive effects in combination with MMV688845. For rifampicin, a FICI of 0.5 was obtained in MHII medium, which is indicative for synergy with MMV688845, leading to a 3- to 4-fold dose reduction. Since rifamycins and MMV688845 do not compete for the same binding site, additive effects were expected, underlining the suitability of MMV688845 as a combination partner.

**Pharmacokinetic properties of MMV688845 *in vivo*.** The pharmacokinetic behavior of MMV688845 was investigated in mice to determine whether the plasma concentrations necessary for efficacy are achieved. The results are shown in Fig. 4. Following oral administration, at 25 mg/kg, exposure was very low, in accordance with an earlier study (9). Given the average peak plasma concentration of 0.07 $\mu$M, higher doses are unlikely to achieve the MIC (6.6 $\mu$M) against strain K21, which was used for *in vivo* efficacy studies.

## DISCUSSION

The data obtained in this study underline the suitability of MMV688845 as a lead structure for drug development against *M. abscessus*. MMV688845 is active *in vitro* against all *M. abscessus* subspecies and clinical isolates analyzed in this study. Thus, we conclude that MMV688845 is active against a broad variety of *M. abscessus* strains. The

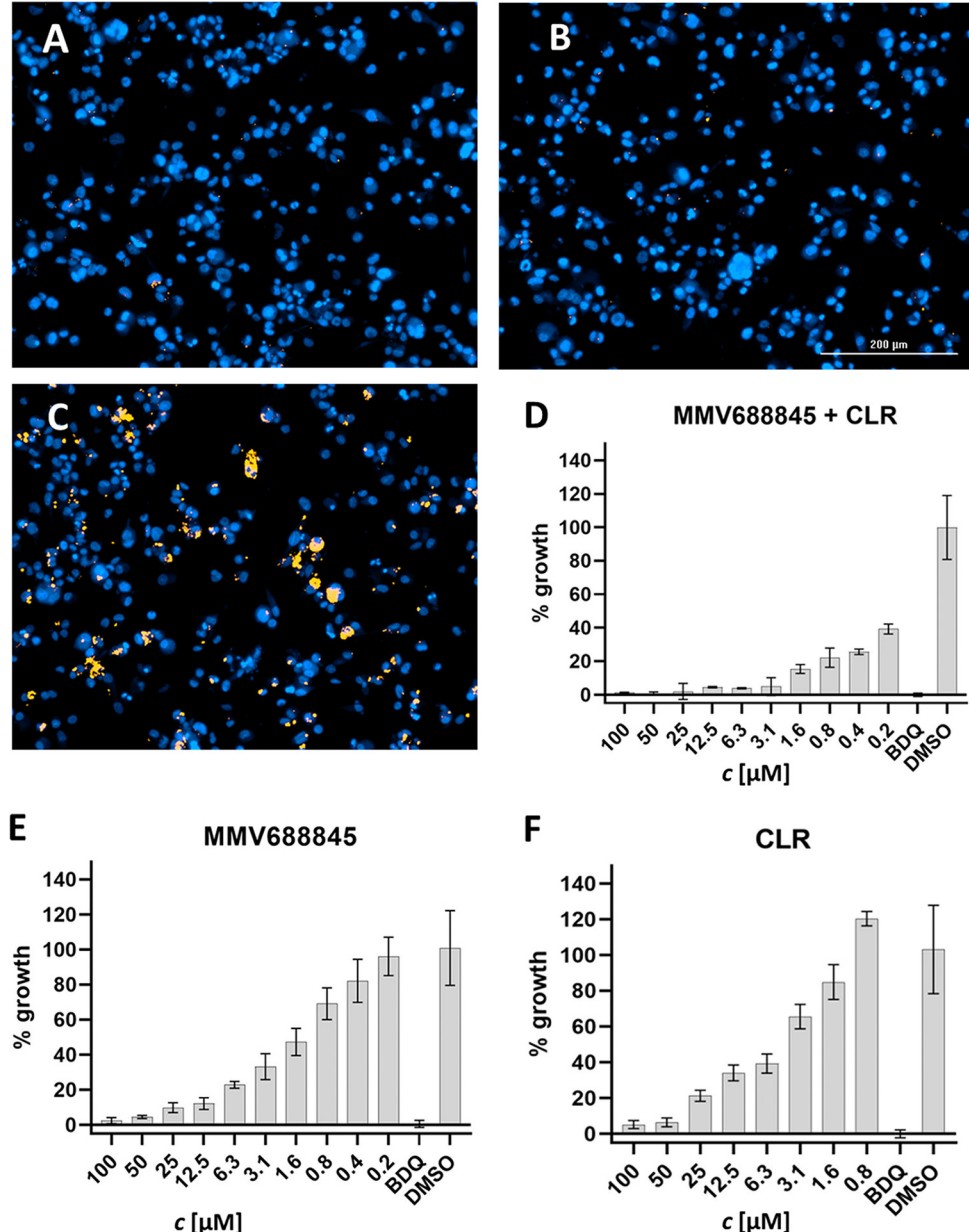

**FIG 3** Intracellular synergy of MMV688845 and clarithromycin (CLR). (A to C) Results of the macrophage infection assay with 1.5 $\mu$M bedaquiline (BDQ) (A), 3.1 $\mu$M CLR plus 1.6 $\mu$M MMV688845 (B), or 1% DMSO (C). Pictures were taken with a Cytation 5 imaging reader fluorescence microscope (BioTek). THP-1 macrophages are shown in blue, and *M. abscessus* (pTEC27) cells are in red. (D to F) growth inhibition of intracellular *M. abscessus* in the presence of 1.6 $\mu$M MMV688845 plus CLR (D), MMV688845 (E), or CLR. Experiments were carried out in duplicate, and the results presented are means, with the standard deviations displayed as error bars.

*in vitro* activity was validated in different media and intracellularly, showing that the lead's structure inhibits the growth of *M. abscessus* independently of the medium composition. By isolation of resistant mutants, MMV688845 was shown to target RNA polymerase in *M. abscessus*. This enzyme is a very well-validated antimycobacterial target and one of the cornerstones of TB therapy and could also be exploited for NTM treatment by using the substance class described here. Particularly noteworthy is that MMV688845 shows pronounced bactericidal activity, which is rare for anti-*M. abscessus* antibiotics (27). The reductions in bacterial counts in broth culture and inside macrophages underscore the idea that MMV688845 or analogues could be an important

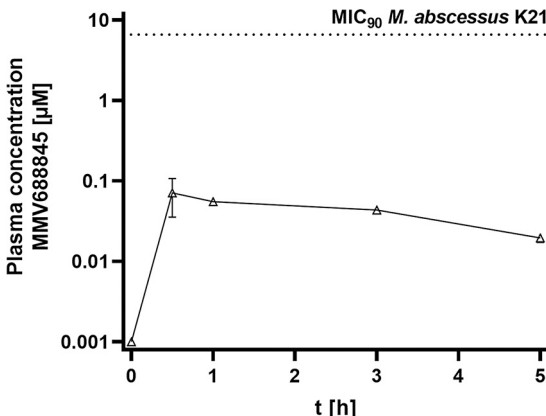

**FIG 4** Plasma concentration-time profile of MMV688845 following a single oral dose of 25 mg/kg in CD-1 mice. The MIC of MMV688845 against *M. abscessus* K21 (Table 1) is indicated by a dotted line. Experiments were carried out in duplicate, and the results presented are as means, with the standard deviations displayed as error bars.

component of a sterilizing *M. abscessus* therapy. The microbiological profile of MMV688845 *in vitro* and in the macrophage infection model has been extended and confirmed as positive in this study. Because mycobacterial infections are treated by combination therapy, the interaction between antibiotics should be analyzed in an early stage of the drug development process. Only a well-tuned combination has the potential to eradicate pathogens like *M. abscessus* while preventing resistance development. Our results show that MMV688845 acts synergistically with the macrolides and ketolides investigated in this study against *M. abscessus* ATCC 19977. The resulting dual inhibition of mRNA biosynthesis and bacterial ribosomal translation is particularly effective. Since *M. abscessus* ATCC 19977 is an *erm41*-positive strain, the results are consistent with the observation of Aziz et al. (16) showing that the RNA polymerase inhibitor rifabutin suppresses inducible macrolide resistance and acts synergistically in *erm41*-positive strains in combination with clarithromycin. We were able to demonstrate synergistic effects of MMV688845 with clarithromycin, in the standard assay in MHII and 7H9 and also in the macrophage infection model. An useful addition in future experiments could be a longer incubation period for MIC determination. This would have the advantage of taking (slowly) inducible resistance into account in the assay, but it would require sufficient stability of the compounds tested.

We observed additive effects for the combinations of MMV688845 with rifabutin and with rifampicin in different assay systems, indicating that the rifamycins do not compete with MMV688845 for one binding site but address different domains in the enzyme, leading to the result mentioned above. X-ray data of the *M. tuberculosis* RNA polymerase/inhibitor complex published by Lin et al. (12) explain this observation, as both binding sites were identified, the observation that the simultaneous administration of MMV688845 and rifamycins, would show an additive effect was expected, because Lin et al. had shown an additive effect for the combination of rifampicin and *Nα*-aroyl-*N*-aryl-phenylalaninamides before against *M. tuberculosis* (12). For other classes of antibiotics, additive effects were observed and antagonism was not found, indicating that combination therapies with MMV688845 and the clinically used antibiotics are possible. Our data support the further development of synthetic RNA polymerase inhibitors as an antimycobacterial compound class. The effects between MMV688845 and antibiotics observed in this study, especially in the synergistic combinations, suggest that clinically required concentrations can be more easily achieved *in vivo* in such combinations. This can lead to reduced drug doses, which might reduce the adverse effects of antibiotic therapy. The synergy shown with clarithromycin suggests that MMV688845 may maintain the efficacy of macrolides during clinical therapy

by repressing transcription of the *erm41* gene if sufficient exposures of MMV688845 or an analogue could be achieved.

However, an *in vivo* study on the bioavailability of MMV688845 found that the compound did not reach sufficient plasma concentrations after oral administration in mice to investigate efficacy. Because of the promising *in vitro* properties of the compound revealed in this study, chemical derivatization is desirable. Based on the results of this study, efforts to improve bioavailability and profile efficacy are in progress, as the mechanism of action of MMV688845 has the potential to exhibit *in vivo* activity against *M. abscessus*. We assume a low metabolic stability for MMV688845, since the amide bonds in the molecule are prone to enzymatic hydrolysis. In this context, derivatives in which the amide bonds in the molecule are shielded against hydrolytic attack might be an interesting option. We highlight that MMV688845 is a promising substance for the development of new antibiotics targeting *M. abscessus*. In this context, it seems to us to be a meaningful approach to improve antibacterial efficacy and oral bioavailability to identify an oral lead compound with proven *in vivo* efficacy.

## MATERIALS AND METHODS

**$MIC_{90}$ determination in 7H9 and MHII by RFP measurement.** MICs were determined against *M. abscessus* ATCC 19977(pTEC27) by the broth microdilution method in 7H9 containing 10% filter sterilized solution of 0.8% sodium chloride, 5.0% bovine serum albumin, and 2.0% dextrose in purified water (ADS) and 0.05% Tween 80 or MHII containing 0.05% Tween 80. A nine-point 2-fold serial dilution of each compound was prepared in 96-well flat-bottom plates (Sarstedt, 3924500). Column 1 of the 96-well plate included eight negative controls (1% dimethyl sulfoxide [DMSO]), and column 2 contained eight positive controls (100 $\mu$M amikacin). Column 12 contained only medium as sterile control. The concentration of the inoculum was $5 \times 10^5$ cells/mL ($OD_{600}$, 0.1 [$1 \times 10^8$ CFU/mL]). The starting inoculum was diluted from a preculture at the mid-log phase ($OD_{600}$, 0.2 to 0.8). The plates were sealed with Parafilm, placed in a container with moist tissue, and incubated for 3 days at 37°C. After incubation, the plates were monitored by RFP measurement at 590 nm (BMG Labtech Fluostar Optima microplate reader). Growth inhibition of >90% is considered to indicate activity. The assay was performed in quadruplicate on each 96-well plate, which was performed in duplicate (8).

**Checkerboard titration assay in 7H9 or MHII and intracellular checkerboard assay.** For each determination, eight concentrations of MMV688845 (50 $\mu$M to 0.039 $\mu$M) were combined with nine concentrations of another substance. The concentration range of the second substance was selected depending on its MIC alone. In total, each compound pair was tested in 72 different concentration combinations. The assay was carried out using the microdilution method as previously described. The assay was performed in two replicates.

**FICI determination.** The fractional inhibitory concentration index (FICI) was used to analyze the results of the checkerboard assay. The FIC was determined as follows: (concentration of drug A in combination/concentration of drug A when used alone) + (concentration of drug B in combination/concentration of drug B when used alone) (26). The FICI was then calculated using the FICs corresponding to the wells with the lowest concentration resulting in >90% inhibition. The lowest FICI of all the nonturbid wells along the turbidity/nonturbidity interface was used (28). Synergy is defined as a FICI of ≤0.5, additivity is defined as a FICI of >0.5 and ≤4, and antagonism is defined as a FICI of >4 (26).

***M. abscessus* infection assay in human macrophages.** For the infection assay, an *M. abscessus* pTEC27 culture ($OD_{600}$, 0.2 to 0.8; mid-log phase) was centrifuged (4,000 rpm, room temperature [RT], 7 min), washed with 7H9 medium with 0.05% Tween 80 (about 10 mL), and vortexed. After another centrifugation (4,000 rpm, RT, 7 min), 7H9 medium was replaced by RPMI medium (at the same volume or a little less to concentrate the bacteria), and the sample was vortexed and incubated at RT for 5 min. After incubation, the bacterial suspension was filtered through a 5-$\mu$m-pore-size filter to remove the clumps. The $OD_{600}$ was determined after filtration (an $OD_{600}$ of 0.1 is equal to $1 \times 10^8$ CFU/mL). The appropriate number of bacteria was incubated in the presence of 10% human serum at 37°C for 30 min for opsonization. A suspension of THP-1 cells ($1 \times 10^6$ cells/mL) in RPMI incomplete medium was incubated with the opsonized *M. abscessus* single-cell suspension (multiplicity of infection [MOI], 5:1) and phorbol myristate acetate (PMA) (40 ng/mL) for 4 h at 37°C under constant agitation. After infection, the THP-1 cell suspension was centrifuged (750 rpm, RT, 10 min) and washed with RPMI medium. Afterward, the cell suspension was dispensed in 96-well plates (Sarstedt, 3924) with $1 \times 10^5$ THP-1 cells/well. The test compounds at the appropriate concentration were added, and the plates, which were sealed with Parafilm, were incubated for 4 days (37°C, 5% $CO_2$). After incubation, the cells were fixed with paraformaldehyde (PFA; 4% in phosphate-buffered saline buffer) for 30 min. After removal of the PFA the cells were stained with ready-made DAPI solution (Sigma, MBD0015). The plates were washed twice with RPMI medium. Image acquisition (DAPI, 386 to 23 nm; RFP, 560 to 25 nm; bright field) and analysis were done with a Cytation 5 imaging reader fluorescence microscope (BioTek). The sum of the spot area (ObjectSumArea) of the RFP channel was used for the calculation of growth inhibition (19).

**$MIC_{90}$ determination in human macrophages.** MICs for *M. abscessus* ATCC 19977(pTEC27) ($OD_{600}$, 0.2 to 0.8, mid-log phase) were determined by the microdilution method in RPMI incomplete medium (RPMI 1640 medium supplemented with 5% fetal bovine serum [FBS], 2% glutamine, and 1%

nonessential amino acids) with 70 $\mu$M amikacin. A 10-point 2-fold serial dilution of each compound was prepared in tissue culture-treated 96-well flat-bottom plates (Sarstedt, 3924). Column 1 of the 96-well plate included eight negative controls (1% DMSO), and column 2 had eight positive controls (1.5 $\mu$M bedaquiline). The plates inoculated with the infected cells (1 $\times$ 10$^5$ THP-1 cells/well) were incubated for 4 days (37°C, 5% CO$_2$). After incubation, the cells were fixed with paraformaldehyde, stained, and washed with RPMI medium. Image acquisition (DAPI, 386 to 23 nm; RFP, 560 to 25 nm; bright field) and analysis were done with a Cytation 5 imaging reader fluorescence microscope (BioTek). The sum of the spot area (ObjectSumArea) of the RFP channel was used for the calculation of growth inhibition (19). The assay was performed in quadruplicate on each 96-well plate, which was performed in duplicate.

**MIC calculation method A.** MIC calculation method A (for the assay described above, including MIC$_{90}$ determination in 7H9 and MHII by RFP measurement, checkerboard assay, and MIC$_{90}$ determination in human macrophages) was done as follows. Every assay plate contained eight wells with DMSO (1%) as a negative control, which correspond to 100% bacterial growth, and eight wells with amikacin (100 $\mu$M) or bedaquiline (1.5 $\mu$M) as a positive control, in which 100% inhibition of bacterial growth was reached. The controls were used to monitor assay quality through the determination of the $Z$ score and for normalizing the data on a plate basis. The $Z$ factor was determined using the following formula: $1 - [3(\text{SD}_{\text{positive control}} + \text{SD}_{\text{DMSO}})/(\text{M}_{\text{positive control}} - \text{M}_{\text{DMSO}})]$, where SD is the standard deviation and M is the mean. Percent inhibition was calculated as follows: $-100 \times [(\text{signal}_{\text{sample}} - \text{signal}_{\text{DMSO}})/(\text{signal}_{\text{DMSO}} - \text{signal}_{\text{positive control}})]$.

For MIC$_{90}$ determination, growth inhibition curves were calculated with OriginPro 2019 software (OriginLab Corporation). The curves were fitted using the following formula: $y = A2 + \{(A1 - A2)/[1 + (x/x_0)^p]\}$, where $A1$ is the initial value for $y$, $A2$ is the final value for $y$, $x$ is the concentration of the test compound, $x_0$ is the concentration of the test compound at the center of the curve, and $p$ is the power.

**MBC determination method A.** For MBC determination *M. abscessus* ATCC 19977 was incubated in a microplate macrophage infection dilution assay for 4 days as described below. Subsequently, the MBC was determined by CFU counting. For this purpose, 6-well plates were used, each filled with 4 mL 7H10 agar supplemented with 0.5% glycerol, 10% ADS, and 400 $\mu$g/mL hygromycin. From the wells where growth inhibition was detected in the microplate dilution assay, 10 $\mu$L (undiluted or diluted 1:100) was plated into one well of the 6-well plates. The colonies were counted after 4 days of incubation at 37°C, and the experiment was carried out in triplicate. Based on the result, the concentration of CFUs per milliliter was calculated. The number of CFUs was also determined in the inoculum prior to the 4-day incubation.

**Bacterial cells and culture media.** *M. abscessus* expressing RFP tdTomato was used for the activity assays. Stocks of the bacteria grown in complete 7H9 broth were stored in approximately 15% glycerol at $-80$°C. Using an inoculation loop, bacteria were spread on 7H10 plates (containing 500 $\mu$g/mL hygromycin) and grown for 5 days in an incubator at 37°C.

Bacteria were grown in complete 7H9 broth supplemented with 10% ADS and 0.05% Tween 80 or in MHII broth supplemented with 0.05% Tween 80. After one colony was scraped off the 7H10 plate, hygromycin (400 $\mu$g/mL) was added for *M. abscessus*(pTEC27) growth. The culture volume was 10 mL in a 50-mL Falcon tube. The tubes were covered to protect the photosensitive hygromycin and shaken in an incubator at 37°C. Solid cultures were grown on 7H10 medium supplemented with 0.5% glycerol and 10% ADS. For *M. abscessus* pTEC27, hygromycin (500 $\mu$g/mL) was added. (ADS supplement is a filter-sterilized solution of 0.8% sodium chloride, 5.0% bovine serum albumin, and 2.0% dextrose in purified water).

**THP-1 cells and culture media.** The THP-1 cells (ATCC TIB-202) used were derived from human monocytes obtained from a 1-year-old male infant with acute monocytic leukemia. The cells were put in 90% FBS–10% DMSO and stored in liquid nitrogen. THP-1 cells were grown in a complete RPMI medium. The cells were grown in a tissue culture flask with a minimum volume of 20 mL and a maximum volume of 50 mL and were incubated in an atmosphere of 95% air and 5% carbon dioxide (CO$_2$) at a temperature of 37°C. The cell density was kept between 0.25 million and 1 million cells/mL. Every 2 or 3 days, the cells were counted and diluted to 0.25 million cells/mL. The cells doubled every 48 h. A culture from nitrogen stock could be subcultured for up to 3 months; after this time, a change in morphology and growth behavior was observed. For culturing of THP-1 cells, RPMI 1640 medium supplemented with 5% FBS, 2% glutamine, and 1% nonessential amino acids.

**Bacterial strains, culture media, and drugs.** *M. abscessus* Bamboo was isolated from the sputum of a patient with amyotrophic lateral sclerosis and bronchiectasis and was provided by Wei Chang Huang, Taichung Veterans General Hospital, Taichung, Taiwan. *M. abscessus* Bamboo whole-genome sequencing showed that the strain belongs to *M. abscessus* subsp. *abscessus* and harbors an inactive clarithromycin-sensitive *erm41* C28 sequevar (29, 30). *Mycobacterium abscessus* subsp. abscessus strain ATCC 19977, harboring the inducible clarithromycin resistance-conferring *erm41* T28 sequevar (31), was purchased from the American Type Culture Collection (ATCC). *Mycobacterium abscessus* subsp. *bolletii* CCUG 50184T, harboring the inducible clarithromycin resistance-conferring *erm41* T28 sequevar (32), and *Mycobacterium abscessus* subsp. *massiliense* CCUG 48898T, harboring the nonfunctional *erm41* deletion sequevar (33), were purchased from the Culture Collection University of Goteborg (CCUG). Clinical isolates covering the *M. abscessus* complex (M9, M199, M337, M404, M422, M232, M506, and M111) were provided by Jeanette W. P. Teo (Department of Laboratory Medicine, National University Hospital, Singapore). For information on the origin of the isolates, see reference 21. The subspecies and *erm41* sequevars of these isolates were determined previously (21). *M. abscessus* subsp. *abscessus* K21 was isolated from a patient and provided by Sung Jae Shin (Department of Microbiology, Yonsei University College of Medicine, Seoul, South Korea) and Won-Jung Koh (Division of Pulmonary and Critical Care Medicine, Samsung Medical Center, Seoul, South Korea) (34). This strain harbors the inactive, clarithromycin-sensitive *erm41* C28 sequevar as determined previously (34). For general bacteria culturing and certain MIC experiments, Middlebrook 7H9 broth (BD Difco) was supplemented with 0.5% albumin, 0.2% glucose, 0.085% sodium

chloride, 0.0003% catalase, 0.2% glycerol, and 0.05% Tween 80. Unless otherwise stated, solid cultures were grown on Middlebrook 7H10 agar (BD Difco) supplemented with 0.5% albumin, 0.2% glucose, 0.085% sodium chloride, 0.5% glycerol, 0.0003% catalase, and 0.006% oleic acid. All drugs were prepared as 10 mM stocks in 100% DMSO.

**Selection of spontaneous resistant mutants.** Spontaneous resistant mutants were selected as described previously (35). Exponentially growing *M. abscessus* Bamboo culture ($10^7$ to $10^9$ CFU) was plated on 7H10 agar containing 50 $\mu$M MMV688845. The plates were incubated for 7 days at 37°C. Apparently resistant colonies were purified and confirmed by restreaking on agar containing the same drug concentration. Two independent batches of resistant mutants were generated in this manner. Genomic DNA was extracted as described previously using the phenol-chloroform method (36). Sanger sequencing of the RpoB (MAB_3869c) genomic region was performed by Genewiz (Genewiz, Inc., South Plainfield, NJ, USA) using 14 primers as previously described (37).

**MIC assay in 7H9 by OD measurement.** MIC determination by $OD_{600}$ was carried out in 96-well plate format as previously described (9, 21). Ninety-six-well plates were initially set up with 100 $\mu$L of 7H9 per well. For each compound, a 10-point 2-fold dilution series starting at twice the desired highest concentration was dispensed into the 96-well plates using a Tecan D300e digital dispenser, with the DMSO concentration normalized to 2%. A bacterial culture grown to mid-log phase ($OD_{600}$, 0.4 to 0.6) was diluted to an $OD_{600}$ of 0.1 ($1 \times 10^7$ CFU/mL). One hundred microliters of the resulting bacteria suspension was dispensed into the 96-well plates containing compounds to give a final volume of 200 $\mu$L per well with an initial $OD_{600}$ of 0.05 ($5 \times 10^6$ CFU/mL) and a final DMSO concentration of 1%. Final compound concentration ranges were typically 50 to 0.098 $\mu$M or 6.25 to 0.012 $\mu$M but were adjusted to 100 to 0.195 $\mu$M for testing of MMV688845-resistant mutant strains. Untreated control wells were included on each plate that contained a bacterial suspension and 1% DMSO. Plates were sealed with Parafilm, stored in boxes with wet paper towels, and incubated at 37°C with shaking (110 rpm). Plates were incubated for 3 days.

**MIC calculation method B.** To determine growth, $OD_{600}$ was measured using a Tecan Infinite M200 plate reader on day 0 and day 3. Two biological replicates were performed. Clarithromycin was included in each experiment as a positive control. For each well in the 96-well plate, bacterial growth was calculated by subtracting the day 0 $OD_{600}$ value from the day 3 $OD_{600}$ value. For each compound series, the bacterial growth values for the untreated control wells were averaged to give the average drug-free bacterial growth. For compound-containing wells, percent growth was calculated by dividing growth values by the average bacterial growth in drug-free wells for the compound series and multiplying by 100. For each compound series, we plotted percentage growth versus compound concentration. By visual inspection of the dose-response curve, we determined the MIC of a compound as the compound concentrations that would result in 90% growth inhibition. The MIC determination was performed twice with different starter cultures. The MICs shown here are the averaged results of the two biological replicates.

**MBC determination method B.** To determine the MBC, CFU measurement was done for the bacteria suspension at an $OD_{600}$ of 0.1 on day 0 and for each well on day 3. Specifically, serial 10-fold dilutions were prepared in phosphate-buffered saline (Thermo Fisher, 10010023) containing 0.05% Tween 80 (PBS-Tween 80) and plated on 7H10 agar. The $MBC_{90}$, $MBC_{99}$, and $MBC_{99.9}$ were defined as the lowest concentrations of drug that reduced the number of CFUs per milliliter by 10-fold, 100-fold, and 1,000-fold, respectively, relative to the day 0 value.

***M. abscessus* homology model.** The built model was then compared to the model for *M. tuberculosis* RpoB (PDB 5UHE) bound to d-AAP 1 in UCSF Chimera (38, 39), and the mutations observed in the *M. abscessus* model were highlighted.

***In vivo* mouse pharmacokinetics.** All animal experiments were approved by the Center for Discovery and Innovation, Institutional Animal Care and Use Committee, and were conducted in compliance with their guidelines. Female CD-1 mice ($n = 2$) were weighed and received a single dose of MMV688845 orally (p.o.) (25 mg/kg of body weight). The compound was formulated in 5% *N,N*-dimethylacetamide, 60% polyethylene glycol 300, and 35% D5W (5% dextrose in water). Blood samples were serially collected via tail snip from each individual mouse at 0.5, 1, 3, and 5 h postdose. Blood (50 $\mu$L) was collected in capillary Microvette K2EDTA tubes (16.444.100; Sarstedt, Inc.) and kept on ice prior to centrifugation at $1,500 \times g$ for 5 min. The supernatant (plasma) was transferred into a 96-well plate and stored at $-80$°C.

**HPLC-MS analysis.** Liquid chromatography-tandem mass spectrometry (LC-MS/MS) analysis was performed on a Sciex Applied Biosystems Qtrap 6500+ triple-quadrupole mass spectrometer coupled to a Shimadzu Nexera 2 high-pressure liquid chromatography (HPLC) system to quantify each drug in plasma. Neat 1-mg/mL DMSO stocks for MMV688845 were serially diluted in 50:50 acetonitrile (ACN)/water to create standard curves and quality control (QC) spiking solutions. Standards and QCs were created by adding 10 $\mu$L of spiking solutions to 90 $\mu$L of drug-free plasma (CD-1 K2EDTA Mouse; Bioreclamation IVT). Twenty microliters of control, standard, QC, or study sample was added to 200 $\mu$L of ACN-methanol (MeOH; 50:50) protein precipitation solvent containing an internal standard (10 ng/mL verapamil). Extracts were vortexed for 5 min and centrifuged at 4,000 rpm for 5 min. One hundred microliters of supernatant was transferred for HPLC-MS/MS analysis and diluted with 100 $\mu$L of Milli-Q deionized water.

Chromatography was performed on an Agilent Zorbax SB-$C_8$ column (2.1 $\times$ 30 mm; particle size, 3.5 $\mu$m) using a reverse-phase gradient. Milli-Q deionized water with 0.1% formic acid was used for the aqueous mobile phase and 0.1% formic acid in ACN for the organic mobile phase. Multiple-reaction monitoring of precursor/product transitions in electrospray positive-ionization mode was used to quantify the analytes. Data processing was performed using Analyst software (version 1.6.2; Applied Biosystems Sciex).

## ACKNOWLEDGMENTS

We thank Nadine Taudte and Jens-Ulrich Rahfeld for providing and maintaining the biosafety 2 facility. We are grateful to Wei Chang Huang (Taichung Veterans General Hospital, Taichung, Taiwan) for providing *M. abscessus* Bamboo, to Jeanette W. P. Teo (Department of Laboratory Medicine, National University Hospital, Singapore) for providing the collection of *M. abscessus* clinical isolates, to Sung Jae Shin (Department of Microbiology, Yonsei University College of Medicine, Seoul, South Korea), to Won-Jung Koh (Division of Pulmonary and Critical Care Medicine, Samsung Medical Center, Seoul, South Korea) for providing *M. abscessus* K21, and to Chao Chen for generating the *M. abscessus* MMV688845-resistant mutants.

This work was funded by the Deutsche Forschungsgemeinschaft (DFG, German Research Foundation), no. 432291016, and by the National Institute of Allergy and Infectious Diseases of the National Institutes of Health under award number R01AI132374.

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
