## [Reviewer comments · Microbiology Spectrum]

Microbiology Spectrum

***In vitro* profiling of the synthetic RNA-Polymerase inhibitor MMV688845 against *Mycobacterium abscessus*.**

Lea Mann, Uday Ganapathy, Rana Abdelaziz, Markus Lang, Matthew Zimmerman, Veronique Dartois, Thomas Dick, and Adrian Richter

Corresponding Author(s): Adrian Richter, Martin-Luther-University Halle-Wittenberg

Review Timeline:

Submission Date:	July 19, 2022
Editorial Decision:	August 22, 2022
Revision Received:	September 16, 2022
Accepted:	October 20, 2022

Editor: Prabakaran Narayanasamy

Reviewer(s): Disclosure of reviewer identity is with reference to reviewer comments included in decision letter(s). The following individuals involved in review of your submission have agreed to reveal their identity: Sangeeta Tiwari (Reviewer #2)

Transaction Report:

DOI: <https://doi.org/10.1128/spectrum.02760-22>

August 22, 2022

Dr. Adrian Richter
Martin-Luther-University Halle-Wittenberg
Halle (Saale)
Germany

Re: Spectrum02760-22 (*In vitro* and *ex vivo* profiling of the synthetic RNA-Polymerase inhibitor MMV688845 against *Mycobacterium abscessus*.)

Dear Dr. Adrian Richter:

Dear Authors,

Link Not Available

Sincerely,

Prabakaran Narayanasamy

Journals Department
Reviewer comments:

Editor:

The anti-Microbial activity of MMV688845 has previously been reported in several publications. Hence we would like to recommend you to add more ADME/PK data and efficacy study in mice models in the revised manuscript in addition to the suggestions given by other reviewers comments.

Reviewer #1 (Comments for the Author):

The manuscript describes a series of in vitro experiments to further characterize the activity of a non-rifamycin RNA polymerase inhibitor hit compound, MMV688845, against *Mycobacterium abscessus* complex. Prior publications reported the in vitro activity of this compound against *M. abscessus* and *M. tuberculosis* in axenic media and macrophage infection assays and showed that members of this chemical series bind the mycobacterial RNA polymerase at a site distinct from rifamycins and do not exhibit

cross resistance with rifamycins. In this study, the authors extend these prior observations by demonstrating in vitro activity against approximately 10 additional bacterial strains representing each of the 3 subspecies in the complex, demonstrating bactericidal activity comparable to rifabutin at a similar multiple of MIC, confirming that MMV688845 selects for resistance-causing mutations in the expected binding site of the *M. abscessus* RpoB protein, demonstrating the lack of cross-resistance with rifabutin (for 1 mutant resistant to each compound), and showing through combination in vitro studies that MMV688845 has synergistic interactions with macrolides and additive interactions with other anti-mycobacterial drugs, including rifamycins. The authors conclude that MMV688845 is an attractive starting point for further hit-to-lead development to improve potency and to identify an orally bioavailable lead with demonstrable in vivo efficacy.

Although the work is technically sound and rigorous, these are rather incremental advances for a hit compound and chemical series that has previously been described but lacks evidence of suitable safety, ADME and PK data to indicate its potential to become an orally available drug. Additional major and minor comments are provided below.

Major comments

The term "ex vivo" is used in the title and manuscript text, seemingly referring to the experiments conducted in an immortalized macrophage-like cell line. This is not an appropriate use of the term.

The methods describe performance of 2-4 replicates for various assays. Such rigor may be an asset to the study. However, it is unclear how the results of the various replicates were incorporated into the results presented in the manuscript. Are the values presented in the text, tables and figures modes, medians, means or some other representation of the replicates?

Showing the structures of marketed antibiotics is unnecessary and the structure of MMV688845 has been published several times. As such, Fig 3 is unnecessary.

A limitation of the work is that all MIC and time-kill assays were carried out for only 3 days, which does not allow sufficient time for induction of known mechanisms causing inducible resistance to macrolides, aminoglycosides, and rifamycins. This should be mentioned in the discussion.

The provenance of the clinical isolates should be described briefly. Were they all isolated from patients in Singapore? Did all patients meet criteria for NTM lung disease? Are there any phylogenetic data to confirm that they are not closely related?

Minor comments

The introduction describes the objective of the synergy testing but none of the other experiments reported.

It is unnecessarily redundant to describe the calculation of the FIC_i in both the results and the methods sections.

Lines 38-9: as written, this sentence could suggest to the reader that the anti-*M. abscessus* activity of MMV688845 is being reported for the first time when in fact it was previously reported in several publications. It should be revised to indicate only that the activity of MMV688845 was characterized further. The authors might also replace "non-tuberculous mycobacteria" with "*M. abscessus* complex" since no other NTMs were studied in this work.

Line 60: the discovery of the compound's anti-*M. abscessus* activity was also reported in PMID 29698397, which should be referenced here.

Line 93: Table 2 is not cited in the text and could be cited at the end of this line.

Line 99: RFP should be spelled out with first use

Line 111: ATCC19977

Line 112: suggest to replace "dose" with "concentration"

Line 116: suggest to replace "16" with "{less than or equal to}16" since concentration range was truncated at 16.

Line 151: unclear what is meant by "complementary" here.

Line 161: contrary to this statement, BlaMab is not responsible for intrinsic resistance to cefoxitin. In the cited reference 14, cefoxitin MICs against the BlaMab knockout and the wild type strain are the same.

Line 213: suggest to replace "higher" with "more potent"

Lines 214-7: please specify which antibiotics this sentence is referring to since FIC_i is reported for CLR and some other drugs

Line 221: Table 5 shows a FIC_i of 0.5, not 0.6. Please reconcile.

Line 231: suggest to delete "against NTMs, especially" since *M. abscessus* was the only NTM studied in this work

Lines 271-3: in this reviewer's opinion, this statement regarding clinical potential of MMV68845 is overly speculative given the lack of any ADME or PK data on the compound to indicate that the exposures needed to produce these effects could be achieved clinically. The statement should be omitted or should be qualified by another phrase like "if sufficient exposures of MMV68845 or an analogue could be achieved clinically..."

Lines 298-307: it should be stated explicitly how long the bacteria were incubated with the MMV68845 in the MBC assay. The text describes incubation periods of 3 and 4 days.

Line 312: "...with >90% inhibition."

Line 314: here the use of the term "indifference" contrasts with the use of "additive" or "additivity" elsewhere in the text (including line 191) when describing the method and interpreting the results. The terminology should be harmonized throughout the manuscript.

Table 1: it should be made clear which MIC values were determined in MHII and which were determined in 7H9. It should also be made clear which MIC values were determined using OD as the readout and which were determined using fluorescence. It should also be made clear in the table, the footnote and/or the text whether the MICs shown are the mean, median, mode or some other measure of the multiple replicates that were performed. Same comment for other tables showing MIC and MBC values.

Table 2: footnote "a" used in the MIC column should not apply to the macrophage infection model results

Table 4: what is the difference between an MIC₉₀ shown as >500 or >125 and footnote "c" denoting no growth inhibition? Also, the results for MMV68845 and RFB appear to be repeated from Table 1, which is unnecessarily duplicative.

Figure 4: please describe whether the y axis in panel D is showing CFU/mL as an absolute number or as a fraction of the DMSO control. Was inhibition measured by CFU counting or by fluorescence? If the latter, then CFU/mL would not be an appropriate y axis label.

Reviewer #2 (Comments for the Author):

The manuscript entitled "In vitro and ex vivo profiling of the synthetic RNA-Polymerase inhibitor 1 MMV688845 against *Mycobacterium abscessus*" by Mann et al. is based on the invitro and ex vivo studies that underlines the attractiveness of synthetic compound MMV688845 as a possible and successful combination therapy against *M. abscessus*. Though the manuscript identifies a potential compound for treatment against *M. abscessus*, a difficult to treat opputunic pathogen, it has several weaknesses experimentally, statistically and in writing that need to be addressed before publication in this Journal. For the improvement of the current state of the manuscript authors should address following major and minor comments mentioned below:

Major comments:

1. Legends for the Figures are missing in some figures. Moreover, figure and table legends should be self-explanatory and need to be cited properly throughout the manuscript, as in its present state it is cumbersome for the reader. E.g., line no. 93, 99, 102, 112, 117, 118, 204 etc.
2. In Figure 1, 4th panel what does authors means by intracellular?
3. Authors should also plot CFU/ml for CLR alone and MMV688845 alone along with CLR+ 1.6 μ M MMV688845 to provide clear contribution of each and how it changes in combination with CLR and RFB in in Figure 4.
4. In Figure-4 the authors assessed the growth inhibition of NMV688845 with clarithromycin, authors should also explain why at Figure 4A. they tested with BDQ why not control as clarithromycin.
5. Authors should provide MTT assay data to provide the estimation of cytotoxicity of the compound to the eukaryotic cells.
6. As MIC₉₀ values are increasing from in vitro just *M. abscessus* vs *M.abcessus* in macrophages. In order to determine the potential of compound (MMV688845) for future implications, authors must perform animal experiments and determine efficacy of the compound.
7. There is no description or refernce is provided for RFP vector pTEC27. It expresses RFP under which promoter ?
8. The manuscript needs corrections at several points in terms of grammar, punctuation use, and sentence clarity to improve the readability and conciseness of the manuscript. E.g., Line no. 21-22, 32, 40, 65, 89, 129, 142, 217 etc.
9. Lack of statistical analysis of the data. The authors have graphed the observed CFUs; however, data must be statistically analyzed and presented in manuscript and figure/table legends in order to have clear comprehension and representation.
10. Authors need to elaborate basis of the statements made in Line no. 225-227.

Minor comments:

1. As the author has cited the information related to the antibiotics tested against *M. abscessus* for the synergy experiment, figure 3 can be removed and is not essential.
2. In lines 420-430 authors have mentioned 100ml or 200ml volume in 96-well plate per well, Is that correct?
3. Authors need to modify first statement in the abstract "however, these natural products show poor potency against *M. abscessus* due to bacterial metabolism and are not used clinically" because as per Table to RFB (belongs to group Rifamysins) has better MIC value than investigated compound MMV688845.

Staff Comments:

Preparing Revision Guidelines

Please return the manuscript within 60 days; if you cannot complete the modification within this time period, please contact me. If you do not wish to modify the manuscript and prefer to submit it to another journal, please notify me of your decision immediately so that the manuscript may be formally withdrawn from consideration by Microbiology Spectrum.

The manuscript entitled “In vitro and ex vivo profiling of the synthetic RNA-Polymerase inhibitor 1 MMV688845 against *Mycobacterium abscessus*” by Mann et al. is based on the invitro and ex vivo studies that underlines the attractiveness of synthetic compound MMV688845 as a possible and successful combination therapy against *M. abscessus*. Though the manuscript identifies a potential compound for treatment against *M. abscessus*, a difficult to treat opputunic pathogen, it has several weaknesses experimentally, statistically and in writing that need to be addressed before publication in this Journal. For the improvement of the current state of the manuscript authors should address following major and minor comments mentioned below:

Major comments:

1. Legends for the Figures are missing in some figures. Moreover, figure and table legends should be self-explanatory and need to be cited properly throughout the manuscript, as in its present state it is cumbersome for the reader. E.g., line no. 93, 99, 102, 112, 117, 118, 204 etc.
2. In Figure 1, 4th panel what does authors means by intracellular?
3. Authors should also plot CFU/ml for CLR alone and MMV688845 alone along with CLR+ 1.6 μ M MMV688845 to provide clear contribution of each and how it changes in combination with CLR and RFB in in Figure 4.
4. In Figure-4 the authors assessed the growth inhibition of NMV688845 with clarithromycin, authors should also explain why at Figure 4A. they tested with BDQ why not control as clarithromycin.
5. Authors should provide MTT assay data to provide the estimation of cytotoxicity of the compound to the eukaryotic cells.
6. As MIC90 values are increasing from in vitro just *M. abscessus* vs M.abcessus in macrophages. In order to determine the potential of compound (MMV688845) for future implications, authors must perform animal experiments and determine efficacy of the compound.
7. There is no description or refernce is provided for RFP vector pTEC27. It expresses RFP under which promoter ?
8. The manuscript needs corrections at several points in terms of grammar, punctuation use, and sentence clarity to improve the readability and conciseness of the manuscript. E.g., Line no. 21-22, 32, 40, 65, 89, 129, 142, 217 etc.

9. Lack of statistical analysis of the data. The authors have graphed the observed CFUs; however, data must be statistically analyzed and presented in manuscript and figure/table legends in order to have clear comprehension and representation.
10. Authors need to elaborate basis of the statements made in Line no. 225-227.

Minor comments:

1. As the author has cited the information related to the antibiotics tested against *M. abscessus* for the synergy experiment, figure 3 can be removed and is not essential.
2. In lines 420-430 authors have mentioned 100ml or 200ml volume in 96-well plate per well, Is that correct?
3. Authors need to modify first statement in the abstract “however, these natural products show poor potency against *M. abscessus* due to bacterial metabolism and are not used clinically” because as per Table to RFB (belongs to group Rifamysins) has better MIC value than investigated compound MMV688845.

Editor:

The anti-Microbial activity of MMV688845 has previously been reported in several publications. Hence we would like to recommend you to add more ADME/PK data and efficacy study in mice models in the revised manuscript in addition to the suggestions given by other reviewers comments.

We agree with the editor and reviewer #2 that in vivo experiments add value to the study. Therefore, we analysed the pharmacokinetics of MMV688845 in an in vivo mouse model and incorporated the data into the manuscript. In addition, the following section was added to the text:

Results section:

Pharmacokinetic properties of MMV688845 in vivo

“The pharmacokinetic behaviour of MMV688845 was investigated in mice to determine whether the plasma concentrations necessary for efficacy are achieved. The graphical representation of the results is shown in Figure 4. Following oral administration, at 25 mg/kg, exposure was very low, in accordance with an earlier study (9). Given the average peak plasma concentration of 0.07 μ M, higher doses are unlikely to achieve the MIC of (6.6 μ M) against strain K21 used for in vivo efficacy studies.”

and (Discussion):

“However, an in vivo study of the bioavailability of MMV688845 found that the compound did not reach sufficient plasma concentrations after oral administration in mice to investigate efficacy. Because of the promising in vitro properties of the compound revealed in this study, chemical derivatisation is desirable. Based on the results of this study efforts are in progress to improve bioavailability and profile efficacy as the mechanism of action of MMV688845 has the potential to exhibit in vivo activity against M. abscessus. We assume a low metabolic stability for MMV688845, since the amide bonds in the molecule are prone to enzymatic hydrolysis. In this context derivatives in which the amide bonds in the molecule are shielded against hydrolytic attack might be an interesting option. We would like to highlight that MMV688845 is a promising substance for the development of new antibiotics targeting M. abscessus. In this context, it seems to us to be a meaningful approach to improve antibacterial efficacy and oral bioavailability to identify an oral lead compound with proven in vivo efficacy.”

Due to the lack of stable pharmacokinetics, an in vivo effectivity study does not seem feasible with MMV688845. Nevertheless, the in vitro data demonstrate the value of MMV688845 as a synthetic RNA polymerase inhibitor with activity against various M. abscessus strains.

Improvement of PK properties can be achieved by synthetic derivatisation of the hit compound, but these experiments are beyond the scope of our study.

Reviewer #1 (Comments for the Author):

The manuscript describes a series of in vitro experiments to further characterize the activity of a non-rifamycin RNA polymerase inhibitor hit compound, MMV688845, against Mycobacterium abscessus complex. Prior publications reported the in vitro activity of this compound against M. abscessus and M. tuberculosis in axenic media and macrophage infection assays and showed that members of this chemical series bind the mycobacterial RNA polymerase at a site distinct from rifamycins and do not exhibit cross resistance with rifamycins. In this study, the authors extend these prior observations by demonstrating in vitro activity against approximately 10 additional bacterial strains representing each of the 3 subspecies in the complex, demonstrating bactericidal activity comparable to rifabutin at a similar multiple of MIC, confirming that MMV688845 selects for resistance-causing mutations in the expected binding site of the M. abscessus RpoB protein, demonstrating the lack of cross-resistance with rifabutin (for 1 mutant resistant to each compound), and showing through combination in vitro studies that MMV688845 has synergistic interactions with macrolides and additive interactions with other anti-mycobacterial drugs, including rifamycins. The authors conclude that MMV688845 is an attractive starting point for further hit-to-lead development to improve potency and to identify an orally bioavailable lead with demonstrable in vivo efficacy.

Although the work is technically sound and rigorous, these are rather incremental advances for a hit compound and chemical series that has previously been described but lacks evidence of suitable safety, ADME and PK data to indicate its potential to become an orally available drug. Additional major and minor comments are provided below.

We thank you for the knowledgeable and detailed review of the manuscript.

Major comments

The term "ex vivo" is used in the title and manuscript text, seemingly referring to the experiments conducted in an immortalized macrophage-like cell line. This is not an appropriate use of the term.

Thank you for this important note about the correct use of the term "ex vivo". We changed the manuscript accordingly.

The methods describe performance of 2-4 replicates for various assays. Such rigor may be an asset to the study. However, it is unclear how the results of the various replicates were incorporated into the results presented in the manuscript. Are the values presented in the text, tables and figures modes, medians, means or some other representation of the replicates?

Two different protocols were used in to calculate the MIC values. We have now named these as methods A and B in the method section and have assigned which values were determined with which method in the tables.

For method A, a calculation was carried out using Origin for which the mean values from the 2-4 replicates served as a basis for the calculation. The calculation via Origin including the equation is described in detail in the material and methods section. For method B percentage growth versus compound concentration was plotted and by visual inspection of the dose-response curve, we determined the MIC of a compound as the compound concentrations that would result in 90% growth inhibition. MICs shown in the script are average results out of two biological replicates.

Showing the structures of marketed antibiotics is unnecessary and the structure of MMV688845 has been published several times. As such, Fig 3 is unnecessary.

The figure was deleted from the manuscript according to the comments of both reviewers.

A limitation of the work is that all MIC and time-kill assays were carried out for only 3 days, which does not allow sufficient time for induction of known mechanisms causing inducible resistance to macrolides, aminoglycosides, and rifamycins. This should be mentioned in the discussion.

We incorporated und discussed this information now in the manuscript. "A useful addition in future experiments could be a longer incubation period for MIC determination. This would have the advantage of taking (slowly) inducible resistance into account in the assay, but would require sufficient stability of the compounds tested."

The provenance of the clinical isolates should be described briefly. Were they all isolated from patients in Singapore? Did all patients meet criteria for NTM lung disease? Are there any phylogenetic data to confirm that they are not closely related?

No, not all isolates are from patients in Singapore. Detailed description of the isolate source is mentioned in the references like: Aziz et al. - 2017 DOI: 10.1128/AAC.00155-17. The Mabs strains called M9-M506 are form NUH Singapore, Mabs Bamboo is from Taiwan and Mabs K21 is from South Korea, which is published here: Dick et al. - 2020. DOI: 10.1128/AAC.01943-19. The references are mentioned in the manuscript.

It is unknown if all patients met the criteria for NTM lung disease. The isolates from Singapore represent isolates from patients (de- identified).

No, there are no phylogenetic data which could confirm that.

Minor comments

The introduction describes the objective of the synergy testing but none of the other experiments reported.

The introduction was rewritten and a paragraph was added to describe the objective of the experiments performed:

In this study MMV688845 was characterized in depth as a potent inhibitor against M. abscessus. MMV688845 was tested against a range of clinical isolates to demonstrate that MMV688845 is not only effective against selected laboratory strains. To evaluate bactericidal effects of MMV688845, we performed determinations of CFU determinations, both in vitro and in the macrophage infection model. For target validation in M. abscessus resistant mutants were isolated and sequenced.

It is unnecessarily redundant to describe the calculation of the FIC_i in both the results and the methods sections.

We decided to keep it in the Material and Methods section and removed the information about the FIC_i calculation from the results section.

Lines 38-9: as written, this sentence could suggest to the reader that the anti-M. abscessus activity of MMV688845 is being reported for the first time when in fact it was previously reported in several publications. It should be revised to indicate only that the activity of MMV688845 was characterized further. The authors might also replace "non-tuberculous mycobacteria" with "M. abscessus complex" since no other NTMs were studied in this work.

We thank you for the important advice and have now presented this information more clearly in the manuscript: "With the lead compound MMV688845, a substance active against M. abscessus complex was characterised in depth." (We had assumed that the citation of the literature took this fact into account, but this was obviously not presented clearly enough in the manuscript.)

Line 60: the discovery of the compound's anti-M. abscessus activity was also reported in PMID 29698397, which should be referenced here.

We added Jeong et al.- 2018 to manuscript (Reference No.10).

Line 93: Table 2 is not cited in the text and could be cited at the end of this line.

Done: "To further mimic infections in human macrophages, a macrophage infection model based on THP-1 cells was used to determine the effect of MMV688845 on bacteria growing inside cells, as activity of MMV688845 in a macrophage model has been reported previously (19) (Table 2)."

Line 99: RFP should be spelled out with first use

Done.

Line 111: ATCC19977

Corrected according to the comment.

Line 112: suggest to replace "dose" with "concentration"

Phrase was changed according to the comment of the reviewer.

Line 116: suggest to replace "16" with "{less than or equal to}16" since concentration range was truncated at 16.

The presentation of data was changed according to the comment.

Line 151: unclear what is meant by "complementary" here.

We routinely used both media to take media dependent effects into account. We changed the sentence to: "For this purpose, two assay media 7H9 and cation-adjusted Mueller-Hinton broth (MHII) were used to determine a possible media dependence of the results."

Line 161: contrary to this statement, BlaMab is not responsible for intrinsic resistance to cefoxitin. In the cited reference 14, cefoxitin MICs against the BlaMab knockout and the wild type strain are the same.

Thank you for this objection. We decided to delete the statement in the manuscript because the information seems not to be relevant for our manuscript.

Line 213: suggest to replace "higher" with "more potent"

Ok, done.

Lines 214-7: please specify which antibiotics this sentence is referring to since FIC_i is reported for CLR and some other drugs

Its referring to azithromycin and telithromycin. We changed the sentence to: "Growth inhibition of 90% or more by azithromycin and telithromycin was not achieved in macrophage infection assay and higher concentrations displayed cytotoxicity against the eukaryotic cells."

Line 221: Table 5 shows a FIC_i of 0.5, not 0.6. Please reconcile.

Corrected according to comment of the reviewer.

Line 231: suggest to delete "against NTMs, especially" since *M. abscessus* was the only NTM studied in this work

Ok, done.

Lines 271-3: in this reviewer's opinion, this statement regarding clinical potential of MMV68845 is overly speculative given the lack of any ADME or PK data on the compound to indicate that the exposures needed to produce these effects could be achieved clinically. The statement should be omitted or should be qualified by another phrase like "if sufficient exposures of MMV68845 or an analogue could be achieved clinically..."

We agree that the statement is (too) speculative and adapted the sentence (Line 300-302) to: The synergy shown with clarithromycin suggests that MMV688845 may maintain the efficacy of macrolides during clinical therapy by repressing transcription of the erm41 gene, if sufficient exposures of MMV68845 or an analogue could be achieved.

Lines 298-307: it should be stated explicitly how long the bacteria were incubated with the MMV68845 in the MBC assay. The text describes incubation periods of 3 and 4 days.

In the MBC determination, we used incubation period of four days. We clarified this in the manuscript.

Line 312: "...with >90% inhibition."

Thank you. We added the word "inhibition".

Line 314: here the use of the term "indifference" contrasts with the use of "additive" or "additivity" elsewhere in the text (including line 191) when describing the method and interpreting the results. The terminology should be harmonized throughout the manuscript.

We decided to consequently use "additive or additivity" and changed the word indifference to additivity in the material and methods section. "The lowest FIC index of all the non-turbid wells along the turbidity/non-turbidity interface was used. Synergy is defined as $FIC_i \leq 0.5$, additivity is defined as $0.5 < FIC_i \leq 4$, and antagonism is defined as $FIC_i > 4$."

Table 1: it should be made clear which MIC values were determined in MHII and which were determined in 7H9. It should also be made clear which MIC values were determined using OD as the readout and which were determined using fluorescence. It should also be made clear in the table, the footnote and/or the text whether the MICs shown are the mean, median, mode or some other measure of the multiple replicates that were performed. Same comment for other tables showing MIC and MBC values.

We have added further footnotes to the table in order to make the respective experimental procedure clearer.

Table 2: footnote "a" used in the MIC column should not apply to the macrophage infection model results

We corrected the mistake in manuscript and changed the footnote accordingly.

Table 4: what is the difference between an MIC90 shown as >500 or >125 and footnote "c" denoting no growth inhibition? Also, the results for MMV688845 and RFB appear to be repeated from Table 1, which is unnecessarily duplicative.

We have decided to replace the values "greater than" with "no growth inhibition" and to remove the values for RFB and MMV from the table in order not to show them again.

Figure 4: please describe whether the y axis in panel D is showing CFU/mL as an absolute number or as a fraction of the DMSO control. Was inhibition measured by CFU counting or by fluorescence? If the latter, then CFU/mL would not be an appropriate y axis label.

We have eliminated an error at this point and corrected the labelling of the y-axis to "growth in %". We have inserted the corrected figure.

Reviewer #2 (Comments for the Author):

The manuscript entitled "In vitro and ex vivo profiling of the synthetic RNA-Polymerase inhibitor 1 MMV688845 against Mycobacterium abscessus" by Mann et al. is based on the invitro and ex vivo studies that underlines the attractiveness of synthetic compound MMV688845 as a possible and successful combination therapy against M. abscessus. Though the manuscript identifies a potential compound for treatment against M. abscessus, a difficult to treat oppotunic pathogen, it has several weaknesses experimentally, statistically and in writing that need to be addressed before publication in this Journal. For the improvement of the current state of the manuscript authors should address following major and minor comments mentioned below:

We thank you for the important comments to improve the manuscript. We have amended the manuscript accordingly and hope to have eliminated the deficiencies/ambiguities.

Major comments:

1. Legends for the Figures are missing in some figures. Moreover, figure and table legends should be self-explanatory and need to be cited properly throughout the manuscript, as in its present state it is cumbersome for the reader. E.g., line no. 93, 99, 102, 112, 117, 118, 204 etc.

We added several citations of figures and tables and hope that the manuscript will now be easier to read and better understandable.

2. In Figure 1, 4th panel what does authors means by intracellular?

Intracellular should refer here to the macrophage infection assay. In this assay method bacteria are incorporated by the phagosome of the immune cells, so they are located intracellular.

3. Authors should also plot CFU/ml for CLR alone and MMV688845 alone along with CLR+ 1.6 μ M MMV688845 to provide clear contribution of each and how it changes in combination with CLR and RFB in in Figure 4.

We have added the graphs for MMV688845 and CLR to the figure for better clarity. (Figure 3)

4. In Figure-4 the authors assessed the growth inhibition of NMV688845 with clarithromycin, authors should also explain why at Figure 4A. they tested with BDQ why not control as clarithromycin.

(Figure 4 is now renamed Figure 3.) The set-up of the macrophage infection model performed in our laboratory is such that BDQ is used as a control substance, because it shows excellent growth inhibition against intracellular M. abscessus.

5. Authors should provide MTT assay data to provide the estimation of cytotoxicity of the compound to the eukaryotic cells.

Cytotoxicity of MMV688845 was investigated in a comparable assay and published (Mann et al. – 2021) DOI: 10.1007/s00726-021-03044-1. We have now clarified this again in the text.

“In a previous study (Mann et al. 2021), the cytotoxicity of MMV688845 was investigated. Cytotoxicity was analysed against five mammalian cell lines including A375 (melanoma), HT29 (colon adenocarcinoma), MCF-7 (breast adenocarcinoma), A2780 (ovarian carcinoma) and non-malignant mouse fibroblasts NIH 3T3. The compound was evaluated using a sulforhodamine B (SRB, Kiton-Red S, ABCR) microculture colorimetric assay in which MMV688845 showed no cytotoxicity (13).”

6. As MIC90 values are increasing from in vitro just M. abscessus vs M.abcessus in macrophages. In order to determine the potential of compound (MMV688845) for future implications, authors must perform animal experiments and determine efficacy of the compound.

We agree with the editor and reviewer #2 that in vivo experiments add value to the study. Therefore, we analysed the pharmacokinetics of MMV688845 in an in vivo mouse model and incorporated the data into the manuscript. In addition, the following section was added to the text:

Results section:

Pharmacokinetic properties of MMV688845 in vivo

“The pharmacokinetic behaviour of MMV688845 was investigated in mice to determine whether the plasma concentrations necessary for efficacy are achieved. The graphical representation of the results is shown in Figure 4. Following oral administration, at 25 mg/kg, exposure was very low, in accordance with an earlier study (9). Given the average peak plasma concentration of 0.07 μ M, higher doses are unlikely to achieve the MIC of (6.6 μ M) against strain K21 used for in vivo efficacy studies.”

and (Discussion):

“However, an in vivo study of the bioavailability of MMV688845 found that the compound did not reach sufficient plasma concentrations after oral administration in mice to investigate efficacy. Because of the promising in vitro properties of the compound revealed in this study,

chemical derivatisation is desirable. Based on the results of this study efforts are in progress to improve bioavailability and profile efficacy as the mechanism of action of MMV688845 has the potential to exhibit in vivo activity against M. abscessus. We assume a low metabolic stability for MMV688845, since the amide bonds in the molecule are prone to enzymatic hydrolysis. In this context derivatives in which the amide bonds in the molecule are shielded against hydrolytic attack might be an interesting option. We would like to highlight that MMV688845 is a promising substance for the development of new antibiotics targeting M. abscessus. In this context, it seems to us to be a meaningful approach to improve antibacterial efficacy and oral bioavailability to identify an oral lead compound with proven in vivo efficacy.”

Due to the lack of stable pharmacokinetics, an in vivo effectivity study does not seem feasible with MMV688845. Nevertheless, the in vitro data demonstrate the value of MMV688845 as a synthetic RNA polymerase inhibitor with activity against various M. abscessus strains. Improvement of PK properties can be achieved by synthetic derivatisation of the hit compound, but these experiments are beyond the scope of our study.

7. There is no description or reference is provided for RFP vector pTEC27. It expresses RFP under which promoter ?

The following reference has now been added to the manuscript: Takaki et al. – 2013, DOI: 10.1038/nprot.2013.068. The RFP vector pTEC27 is commonly used for Mycobacteria. Further information can be found in the reference.

8. The manuscript needs corrections at several points in terms of grammar, punctuation use, and sentence clarity to improve the readability and conciseness of the manuscript. E.g., Line no. 21-22, 32 We have reviewed the entire manuscript and edited it for punctuation and clarity., 40, 65, 89, 129, 142, 217 etc.

We have reviewed the entire manuscript and edited it for punctuation and clarity.

9. Lack of statistical analysis of the data. The authors have graphed the observed CFUs; however, data must be statistically analyzed and presented in manuscript and figure/table legends in order to have clear comprehension and representation.

For CFU data shown in the manuscript, we have performed a one-way ANOVA multiple comparison test using GraphPad Prism 8 software to compare treated groups with the DMSO control. The results are shown in Figure 1. Table 2 refers to these values and the statistical analysis is mentioned in the legend.

10. Authors need to elaborate basis of the statements made in Line no. 225-227.

Since rifamycins and MMV688845 do not compete for the same binding site at the mycobacterial RNA polymerase, additive efficacy was expected, underlining the suitability of MMV688845 as a combination partner. If both would bind to the same binding site, they would compete for binding and an additive effect of the combination is unlikely.

Minor comments:

1. As the author has cited the information related to the antibiotics tested against *M. abscessus* for the synergy experiment, figure 3 can be removed and is not essential.

The figure was deleted from the manuscript as requested by both reviewers

2. In lines 420-430 authors have mentioned 100ml or 200ml volume in 96-well plate per well, Is that correct?

The unit (ml) is a mistake of was corrected to μ l.

3. Authors need to modify first statement in the abstract "however, these natural products show poor potency against *M. abscessus* due to bacterial metabolism and are not used clinically" because as per Table to RFB (belongs to group Rifamysins) has better MIC value than investigated compound MMV688845.

The introduction was rewritten and the statement regarding rifampicin was corrected according to the comment of the reviewer:

*"Due its bactericidal activity, rifampicin is a key drug for the treatment of TB. However, this natural product shows poor potency against *M. abscessus* due to enzymatic modification and its clinical use is limited*

October 20, 2022

Dr. Adrian Richter
Martin-Luther-University Halle-Wittenberg
Halle (Saale)
Germany

Re: Spectrum02760-22R1 (*In vitro* profiling of the synthetic RNA-Polymerase inhibitor MMV688845 against *Mycobacterium abscessus*.)

Dear Dr. Adrian Richter:

Your manuscript has been accepted, and I am forwarding it to the ASM Journals Department for publication. You will be notified when your proofs are ready to be viewed.

Sincerely,

Prabakaran Narayanasamy
Editor, Microbiology Spectrum
